# Quantum chaos in $\mathcal{PT}$-symmetric quantum systems

Kshitij Sharma[1*], Himanshu Sahu [1,2 †] and Subroto Mukerjee[1‡]

**1** Department of Physics, Indian Institute of Science, C.V. Raman Avenue, Bangalore 560012, India.

**2** Department of Instrumentation & Applied Physics, Indian Institute of Science, C.V. Raman Avenue, Bangalore 560012, Karnataka, India.

⋆ kshitijvijay@iisc.ac.in , † himanshusah1@iisc.ac.in , ‡ smukerjee@iisc.ac.in

August 5, 2024

## Abstract

In this study, we explore the interplay between $\mathcal{PT}$-symmetry and quantum chaos in a non-Hermitian dynamical system. We consider an extension of the standard diagnostics of quantum chaos, namely the complex level spacing ratio and out-of-time-ordered correlators (OTOCs), to study the $\mathcal{PT}$-symmetric quantum kicked rotor model. The kicked rotor has long been regarded as a paradigmatic dynamic system to study classical and quantum chaos. By introducing non-Hermiticity in the quantum kicked rotor, we uncover new phases and transitions that are absent in the Hermitian system. From the study of the complex level spacing ratio, we locate three regimes – one which is integrable and $\mathcal{PT}$-symmetry, another which is chaotic with $\mathcal{PT}$-symmetry and a third which is chaotic but with broken $\mathcal{PT}$-symmetry. We find that the complex level spacing ratio can distinguish between all three phases. Since calculations of the OTOC can be related to those of the classical Lyapunov exponent in the semi-classical limit, we investigate its nature in these regimes and at the phase boundaries. In the phases with $\mathcal{PT}$-symmetry, the OTOC exhibits behaviour akin to what is observed in the Hermitian system in both the integrable and chaotic regimes. Moreover, in the $\mathcal{PT}$-symmetry broken phase, the OTOC demonstrates additional exponential growth stemming from the complex nature of the eigenvalue spectrum at later times. We derive the analytical form of the late-time behaviour of the OTOC. By defining a normalized OTOC to mitigate the effects caused by $\mathcal{PT}$-symmetry breaking, we show that the OTOC exhibits singular behaviour at the transition from the $\mathcal{PT}$-symmetric chaotic phase to the $\mathcal{PT}$-symmetry broken, chaotic phase.

# 1   Introduction

Over the years, the notion of chaos in a quantum system has been placed on a firm footing despite the absence of a phase space in which to describe the dynamics, [1, 2]. Diagnostics such as the energy level spacing distribution have been employed to characterize chaotic quantum Hamiltonians due to their similarities with random matrices [3, 4]. Random Matrix Theory (RMT) is a powerful tool that accurately describes the spectral statistics of quantum systems whose classical counterparts exhibit chaotic behaviour. In cases where quantum Hamiltonians correspond to integrable classical systems, the Berry-Tabor conjecture proposes that their level spacing distributions have Poissonian forms, [5]. On the other hand, for quantum Hamiltonians with chaotic classical counterparts, the Bohigas-Giannoni-Schmit conjecture proposes that the level statistics should correspond to one of the three classical ensembles of RMT (Wigner-Dyson ensembles) [6], namely – the Gaussian unitary ensemble (GUE), the Gaussian orthogonal ensemble (GOE), and the Gaussian symplectic ensemble (GSE) that come out of Wigner's surmise, [7, 8]. A similar diagnostic is the energy level spacing ratio, which has proven more versatile due to its independence from local energy densities, [9].

    The classical kicked rotor has proven paradigmatic in understanding chaos classically in time-dependent 1D systems. It can be stroboscopically evolved using the Chirikov standard map [10] and exhibits a transition from integrability to chaos with increasing kicking strength [11]. The quantum version of the classical kicked rotor also displays a transition from being integrable to chaotic [12–14]. This paper explores chaos in a non-Hermitian version of the quantum kicked rotor.

    In recent studies, it has been observed that the out-of-time-order correlator (OTOC) carries information on chaos in quantum systems [15]. The Lyaponuv exponent, which is a classical measure of chaos, can be extracted from taking derivatives of the canonical variables with respect to initial conditions. When such a derivative is expressed as a Poisson bracket, and the brackets are transformed into commutators, one obtains the OTOC [16].

    The out-of-time-order correlator (OTOC) [16–19] is another possible measure of quantum chaos that may be used to identify an analogue of the Lyapunov exponent, providing a connection with classical chaos, e.g., via the butterfly effect. Previously, OTOCs have been calculated in the context of information scrambling [20–23], quantum butterfly effects [24],

many-body localization [25], "fast scrambling" [26–28], dynamical and topological phase transitions [29–32], and open quantum systems [33, 34]. Recently, the experimental implementation of many-body time-reversal protocols in atomic quantum systems has attracted attention for its potential to measure OTOCs experimentally [35–38] leading to several specific experimental proposals to measure OTOCs and also the first experimental demonstrations [15, 39–41]

The OTOC has been calculated for the Hermitian kicked rotor [42]. It has been observed that at initial times, the OTOC exhibits exponential growth in the chaotic regime, which is absent in the integrable regime. Therefore, it has been possible to extract a quantum equivalent of the classical Lyapunov exponent, at least in a semi-classical limit.

Non-Hermitian Hamiltonians have been studied extensively over the last few years due to the discovery of phenomena such as the non-Hermitian skin effect [43], the presence of exceptional points [44] and interesting topological properties in these systems. [45, 46]. Bender showed that if a Hamiltonian commutes with the operator $\mathcal{PT}$ where $\mathcal{P}$ is the (unitary) parity operator and $\mathcal{T}$ is the (anti-unitary) time reversal operator, it can possess a completely real spectrum of eigenvalues without necessarily being Hermitian [47,48]. In fact, one can typically choose a parameter to tune in such a $\mathcal{PT}$ symmetric Hamiltonian to go from a $\mathcal{PT}$ symmetric phase to a $\mathcal{PT}$ symmetry broken phase, in which the energy eigenvalues are complex [47].

It is thus interesting to investigate whether a system can exhibit both an integrable to chaotic transition and a $\mathcal{PT}$ symmetry-breaking transition, and if so, what are the possible resultant phases? Motivated by the above question, in this work, we study a non-Hermitian extension of the kicked rotor model, which we henceforth refer to as the $\mathcal{PT}$ symmetric kicked rotor (PTKR) model.

Since the PTKR model is non-Hermitian and may have complex eigenvalues, we must use generalizations of the previously discussed diagnostics of quantum chaos [49] that apply to complex eigenvalues. For example, we study the complex energy level spacing ratio (CLSR) [50, 51] instead of the conventional real energy level spacing ratio (RLSR) [9]. Additionally, we calculate a suitably defined OTOC for non-Hermitian systems. [52]. Recently, the time evolution of the OTOC has been studied for the PTKR model and its classical counterpart in the vicinity of a $\mathcal{PT}$ symmetry-breaking transition in parameter space. [52–54]. Our study extends this study to include the entire phase diagram of the PTKR model. It provides critical insights into the interplay of the $\mathcal{PT}$ symmetry-breaking transition and the transition from integrability to chaos.

The rest of the paper is structured as follows. In section 2, we introduce the PTKR model. In section 3, we discuss the two diagnostics of quantum chaos – complex level spacing ratio and out-of-ordered correlator. The main results based on these diagnostics are presented in section 4. Finally, we conclude in section 5 discussing our work's implications and future directions.

## 2  Model

We modify the Hermitian quantum kicked rotor model by adding a non-Hermitian term that preserves $\mathcal{PT}$ symmetry. The resultant model is the PTKR model described by the Hamiltonian

$$H = \frac{p^2}{2m} + V(\theta) \sum_n \delta\left(n - \frac{t}{\tau}\right) \tag{1}$$

where

$$V(\theta) = K \frac{(\cos\theta + i\lambda \sin\theta)}{\sqrt{1 + \lambda^2}}. \tag{2}$$

Here $\theta$ is the angle the rotor makes with a pre-defined direction and the momentum $p = -i\hbar d/d\theta$. When set to zero, the non-Hermiticity parameter $\lambda$ gives the Hermitian model. The action of the parity operator $\mathcal{P}$ is defined as follows-

$$\mathcal{P}\hat{\theta}\mathcal{P}^{-1} = -\hat{\theta} \qquad \mathcal{P}\hat{p}\mathcal{P}^{-1} = -\hat{p} \tag{3}$$

The time reversal operator $\mathcal{T}$ is just the complex conjugation operator. Thus, we have that $[H, \mathcal{PT}] = 0$ but $[H, \mathcal{P}] \neq 0$ and $[H, \mathcal{T}] \neq 0$. The PTKR model Hamiltonian is time-dependent and, therefore, cannot be diagonalized by a time-independent transformation. However, since the time dependence is periodic with a constant period $\tau$, we can define a Floquet evolution operator of the time-independent Floquet Hamiltonian $\mathcal{H}_\mathcal{F}$.

The Floquet evolution operator is usually a unitary operator defined over a fixed time period of the system, providing a stroboscopic view of the system's evolution. For our Hamiltonian, the Floquet operator is no longer unitary. We define it as follows

$$\mathcal{F}_{t_0} = T \exp\left\{\frac{-i}{\hbar} \int_{t_0}^{\tau + t_0} H(t)dt\right\} \tag{4}$$

where $T$ stands for time ordering. The form of the operator thus becomes

$$\mathcal{F} = \exp\left\{-i\frac{p^2}{4}\right\} \exp\{-iV(\theta)\} \exp\left\{-i\frac{p^2}{4}\right\} \tag{5}$$

where we have taken $\hbar = 1, \tau = 1$ and $m = 1$. The above-defined operator is time-independent and can be diagonalized to obtain its eigenvalues. To extract the eigenvalues of the PTKR model Hamiltonian from its Floquet evolution operator, we take the logarithm of the operator's eigenvalues and multiply them by $i$. Due to the periodic nature of phases, the eigenvalue spectrum of the Floquet Hamiltonian is folded into a single interval modulo the period $2\pi$. Since this procedure causes eigenvalues that would otherwise have been far apart to now be proximate, it generates chaos.

## 3  Methods

### 3.1  Energy Level Spacing Ratio

A quantity derivable from the eigenvalue spectrum of a random matrix is the level spacing ratio ($r$). For the case of a real energy spectrum, it is defined as

$$r = \frac{1}{N_M - 2} \sum_{\beta=1}^{N_M - 2} \frac{\min(\mu_{\beta+2} - \mu_{\beta+1}, \mu_{\beta+1} - \mu_\beta)}{\max(\mu_{\beta+2} - \mu_{\beta+1}, \mu_{\beta+1} - \mu_\beta)}. \tag{6}$$

In the above equation, $\mu_i$ is the $i^{\text{th}}$ eigenvalue out of a total $N_M$ eigenvalues arranged in ascending order. The advantage $r$ has over calculating the level spacing distribution is that it does not require an unfolding procedure (i.e. a normalization of the level spacing distribution by the local density of states). Quantum chaotic systems possess the same energy level spacing distribution as random Hamiltonians with the same set of symmetries [3].

To extend the analysis of the energy level spacing distribution to non-Hermitian systems, we use the complex level spacing ratio $\xi$ [50] defined as

$$\xi_K = \frac{z_K^{\text{NN}} - z_K}{z_K^{\text{NNN}} - z_K} = r_K \exp\{i\theta_K\} \tag{7}$$

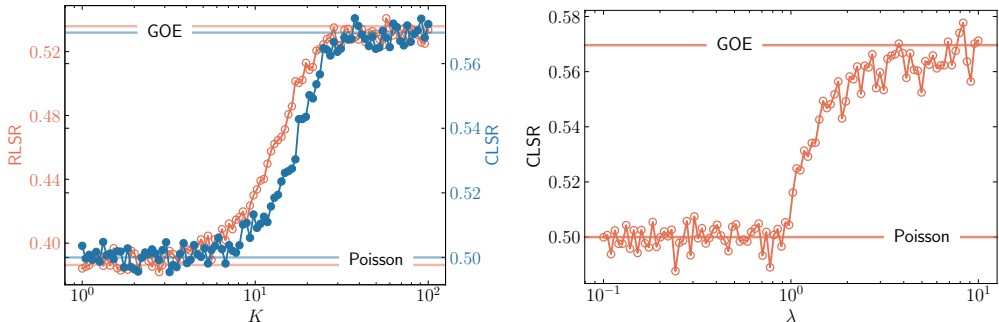

Figure 1: **Left** The CLSR and RLSR as a function of the kicking strength $K$, calculated for $N = 8005$ with $\hbar = 0.2$. Horizontal lines represent analytical values of RLSR for Poisson and GOE statistics. The figure shows that the CLSR is as good an indicator of the transition from integrability to chaos as the more commonly employed RLSR. **Right** The CLSR and RLSR as a function of the non-Hermiticity parameter $\lambda$, calculated for $N = 4095$ with $\hbar = 0.2$ and $K = 0.15$. Horizontal lines represent the values of CLSR for the Poisson and GOE distributions. The figure resembles the transition observed for the CLSR on the left got by varying $K$ with $\lambda = 0$. However, it is obtained by tuning $\lambda$ instead of $K$. This suggests that the system can transition from a $\mathcal{PT}$-symmetric integrable phase to a $\mathcal{PT}$-symmetric chaotic phase with increasing non-Hermiticity while the value of $K$ is kept constant.

The complex level spacing ratio (CLSR) can be defined in two ways: One is by averaging over the magnitudes of $\xi$ to obtain $\langle r \rangle$ (derived from the set of $r_K$ values), and the other is averaging over the angular distribution to obtain $-\langle \cos \theta \rangle$ (derived from the set of $\theta_K$ values). Only $\langle r \rangle$ can be defined in the limit when the spectrum becomes real. However, in this limit, Eq. 7 does not yield the standard real level spacing ratio (RLSR) for a real spectrum. We, henceforth, refer only to the mean value $\langle r \rangle$ as the complex level spacing ratio (CLSR). Calculations of $-\langle \cos \theta \rangle$ can be found in the supplementary material. The universality classes for random matrices that describe Hermitian matrices in terms of Gaussian Ensembles have suitable generalization to complex matrices [49]. We calculate the CLSR for these universality classes, one of which the PTKR model belongs.

There are two computational techniques to improve the statistics in our calculation. The first is to replace $m$ in Eq. (1) by $m + \Delta m_p$, where $\Delta m_p$ is a small random number selected independently for each $p$. This ensures that no underlying symmetries leading to unwanted degeneracies occur. This is important as degeneracies must be avoided when calculating level spacing ratios. The second is to average over a small window of values of the kicking strength $K$. This is done to avoid any accidental resonance in the $K$ values. Although, this averaging wasn't performed while plotting the colormaps to keep the transitions intact.

## 3.2   Out-of-time-order correlator

Recently, the study of OTOCs has extended to non-Hermitian systems, particularly those that possess $\mathcal{PT}$-symmetry. In the context of the QKR, these studies have shown that OTOC increases as a power law in time in the broken $\mathcal{PT}$-symmetry phase [52, 53, 55]. In this study, we investigate the nature of the OTOC in the different phases identified employing the complex level spacing ratio. The OTOC is typically defined as

$$C(t) \equiv -\langle [W(t), V(0)]^2 \rangle, \qquad (8)$$

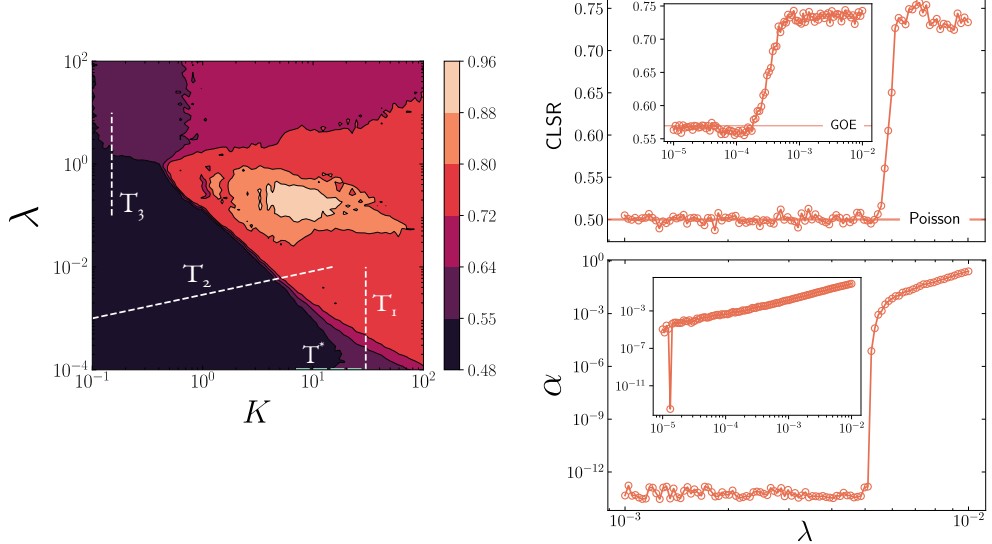

Figure 2: **Left**: The CLSR for varying values of the kicking strength $K$ and the non-Hermiticity $\lambda$, computed for $N = 6005$, $\hbar = 0.2$. The dotted lines T1 and T2 represents the transitions across the $\mathcal{PT}$-symmetric integrable phase to $\mathcal{PT}$ symmetry broken chaotic phase and the $\mathcal{PT}$-symmetric chaotic phase to $\mathcal{PT}$ symmetry broken chaotic phase, respectively. Line T* is the Hermitian transition from the integrable to chaotic phase. T3 marks a transition similar to T* in which $K$ is held constant and $\lambda$ varied. **Center**: The CLSR across the transition **Main panel**: T1 **Inset** T2, calculated for $N = 4095$. The horizontal lines show the standard values of the CLSR for the GOE and Poisson distributions. **Right**: The absolute value of the maximum imaginary part of an energy eigenvalue, $\alpha$, across the transition **Main panel**: T1 **Inset** T2, calculated for $N = 4095$. It can be seen that while $\alpha$ shows an abrupt change along T2, it seems to increase smoothly along T1. The CLSR on the other hand, shows an abrupt transition along both T1 and T2.

where $\langle \cdots \rangle$ represents the expectation values with respect to a state $|\Psi\rangle$. $W(t)$ and $V(t)$ are operators at time $t$ in the Heisenberg representation. In what follows, we choose $W(t) = \hat{p}(t)$, and $V(0) = \hat{p}(0)$ in Eq. (8). The state $|\Psi\rangle$ is a Gaussian wave packet

$$|\Psi\rangle = \sum_{k=-\infty}^{\infty} a_k^{(0)} |k\rangle \tag{9}$$

where

$$a_k^{(0)} \sim \exp\left(-\frac{\hbar_{\text{eff}}^2 (k - k_0)^2}{2\sigma^2}\right), \qquad \hat{p}|k\rangle = \hbar_{\text{eff}} k |k\rangle.$$

We choose $p_0 = \hbar_{\text{eff}} k_0 \in [-\pi, \pi]$ and $\sigma = 4$. Numerically, $|\Psi\rangle$ is represented in a finite basis of eigenstates $|k\rangle$, $k \in [-N; N-1]$. The Hamiltonian is expressed as a matrix in this basis of $|k\rangle$ states, and all calculations are also performed w.r.t this basis. In both, the calculation of the OTOC and CLSR, an "effective value" of the Planck constant $\hbar_{\text{eff}}$ has been employed as a parameter to tune the amount of discreetness of the momentum operator [42]. This allows one to make a connection with the classical model by taking the limit $\hbar_{\text{eff}} \to 0$.

Table 1: The above table consists of data gathered on CLSR and RLSR of standard known values in the integrable regime and random matrix ensemble calculations in the chaotic regime [9, 51].

| Universality Class | RLSR | CLSR |
|---|---|---|
| Poisson | 0.386 | 0.50 |
| GOE | 0.536 | 0.57 |

## 4 Results

We now present the results of the calculations discussed in Sec. 3 for the PTKR model.

### 4.1 CLSR

For the Hermitian case ($\lambda = 0$), the mean RLSR can be used to identify the transition from integrability to chaos. In what follows, we show that the CLSR can also be used for the same purpose and takes on characteristic values in the two regimes.

In Fig. 1, we show the mean RLSR as well as the CLSR with varying values of the kicking strength $K$, for $\hbar = 0.2$ and system-size $N = 8005$. We notice that the transition points are reasonably independent of the value of $N$. Thus, in what follows, the system size is chosen to be large enough so that all quantities are well converged. We find that both the CLSR and RLSR display a transition at the same value of $K$. However, the values in the two regimes are different. Furthermore, the values of the CLSR and RLSR that are obtained by the PTKR model in the chaotic limit of the Hermitian case match with those obtained from averaging over purely random matrices in the Gaussian Orthogonal Ensemble (GOE). The transition in the Hermitian case has been highlighted in the color plot as T* for aid in comparison.

We now consider the non-Hermitian case ($\lambda \neq 0$), which is of particular interest in this study. In Fig. 2, we show the behavior of the CLSR for varying values of kicking strength $K$ and the non-Hermiticity $\lambda$, computed for $\hbar_{\text{eff}} = 0.2$, and system-size $N = 6005$. We define $\alpha = \max\{\text{Im}(E)\}$ which only becomes non-zero when the spectrum starts to possess imaginary eigenvalues of the energy or, equivalently, when $\mathcal{PT}$-symmetry breaks (see supplementary material). We find that the phase diagram derived from the CLSR consists of three regimes: a $\mathcal{PT}$-symmetric integrable phase, a $\mathcal{PT}$-symmetric chaotic phase, and $\mathcal{PT}$-symmetry broken chaotic phase.

There are three possible transitions that can occur: 1) From the $\mathcal{PT}$-symmetric integrable phase to the $\mathcal{PT}$-symmetry broken chaotic phase, which we label $T_1$, 2) from the $\mathcal{PT}$-symmetric chaotic phase to the $\mathcal{PT}$-symmetry broken chaotic phase, which we label $T_2$ and 3) from the $\mathcal{PT}$-symmetric integrable phase to $\mathcal{PT}$-symmetric chaotic phase, which we label $T_3$. These are shown in Fig. 2. $T_3$ is similar to the transition seen in the Hermitian case. This is because, along $T_3$, we see that the CLSR goes from 0.50 to 0.57, which is exactly what is observed when going from the integrable to the chaotic regime for $\lambda = 0$. For transition $T_2$, we see a corresponding abrupt change in the value of $\alpha$ across the transition from the $\mathcal{PT}$-symmetric integrable regime to the $\mathcal{PT}$-symmetry broken chaotic regime. It can be seen that $\mathcal{PT}$-symmetry breaking at the transition $T_1$ does not occur abruptly as can be expected for a system with finite Hilbert space dimension (See Fig. 2). Therefore, the exact position of the transition point cannot be determined with great precision; however, the CLSR does show a clear transition. We define a threshold value of $\alpha$ above which the $\mathcal{PT}$-symmetry is assumed to be broken. The threshold, keeping in mind numerical errors, is taken to be $\alpha \leq 10^{-10}$ for unbroken $\mathcal{PT}$-symmetry. We observe that the CLSR takes on the same value of 0.57 in the

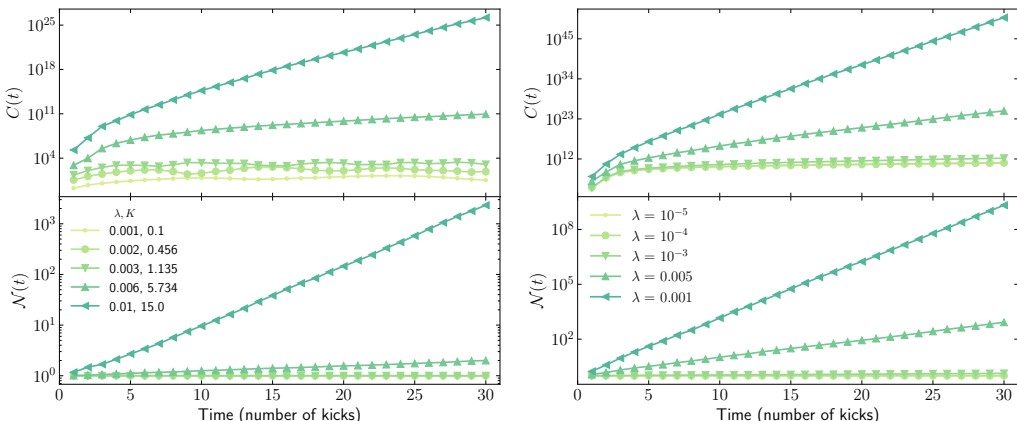

Figure 3: The OTOC $C(t)$ and state norm $\mathcal{N}(t)$ vs $t$ across the transition along **Left** $T_1$ and **Right** $T_2$ shown in Fig. 2, calculated for system size $N = 2^{14}$. For T1 (Left), the values $(\lambda, K) = \{(0.001, 0.1), (0.002, 0.456), (0.003, 1.135)\}$ correspond to the $\mathcal{PT}$-symmetry unbroken integrable phase, while the values $(\lambda, K) = \{(0.006, 5.734), (0.01, 15.0)\}$ correspond to the $\mathcal{PT}$-symmetry broken chaotic phase. For T2 (right), the values $(\lambda, K) = \{(10^{-5}, 30), (10^{-4}, 30), (10^{-3}, 30)\}$ correspond to the $\mathcal{PT}$-symmetry unbroken chaotic phase, while the values $(\lambda, K) = \{(0.005, 30), (0.001, 30.0)\}$ correspond to the $\mathcal{PT}$-symmetry broken chaotic phase. In both cases, the $\mathcal{PT}$-symmetric broken phase displays a late-time exponential growth of the OTOC and the state norm.

$\mathcal{PT}$-symmetric chaotic regime as it does in the chaotic regime of the Hermitian model. However, we find that in the $\mathcal{PT}$-symmetric chaotic regime, the value of $\alpha$ can be greater than the threshold of $10^{-10}$, demonstrating that the highest imaginary value of the eigenenergies is not a particularly good diagnostic to detect the absence of $\mathcal{PT}$-symmetry.

It is interesting to note that the transition from the $\mathcal{PT}$-symmetric integrable phase to the $\mathcal{PT}$-symmetric chaotic also occurs across $T_3$. The hermitian case $\lambda = 0$ exhibits a transition from $\mathcal{PT}$- symmetric integrable to the $\mathcal{PT}$-symmetric chaotic phase as indicated by $T*$. A similar transition can thus be expected when one varies $K$ for $\lambda$ sufficiently small but not equal to zero as seen in the left panel of Fig. 2. The same transition can also be effected by varying $\lambda$ but not $K$ as indicated by $T_3$. This is somewhat surprising as it seems to lack any Hermitian counterpart. One is thus able to induce chaos, as seen from the measured CLSR values, by increasing $\lambda$ whilst holding $K$ to the traditional integrable value. The nature of this transition is intriguing and requires further study, which we defer to future work.

## 4.2 OTOC

We now present our results from the calculation of the OTOC. More specifically, we present the OTOC calculations for various phases obtained from the CLSR analysis and across the transitions $T_1$ and $T_2$.

*$\mathcal{PT}$-symmetric integrable phase*: The Hermitian case $\lambda = 0$ has been studied previously [42], where it was shown that the OTOC follows a power law for all time scales in the integrable phase. On the other hand, the OTOC features a transition from exponential growth at early times $t < t_E$ to a power law in the chaotic phase. In Fig. 3 (right), we show the OTOC $C(t)$ for the $\mathcal{PT}$-symmetric integrable phase with parameters $(\lambda, K) \in \{(0.001, 0.1), (0.002, 0.456), (0.03, 1.135)\}$, and system-size $2^{14}$. We find that the OTOC exhibits a behavior similar to the

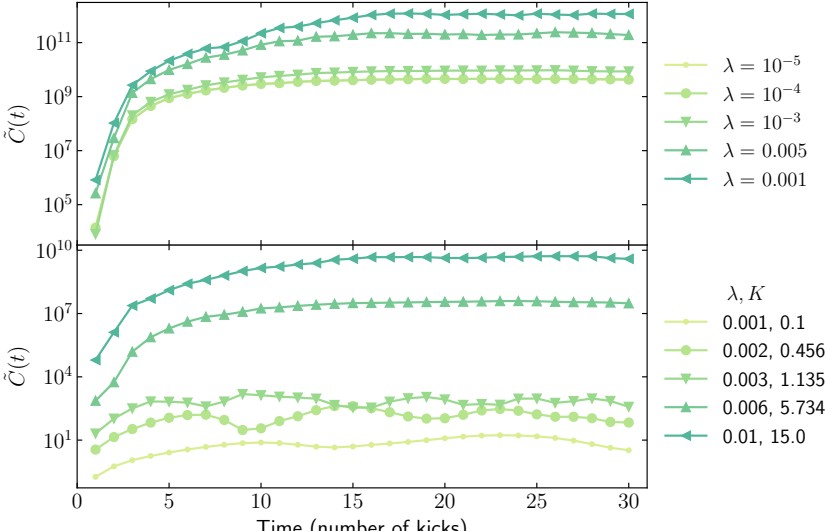

Figure 4: The normalized OTOC $\tilde{C}(t)$ vs $t$ corresponding to the OTOC $C(t)$ shown in Fig. 3. At late times, $\tilde{C}(t)$ has a constant value after accounting for the exponential growth resulting from broken $\mathcal{PT}$-symmetry.

Hermitian case, which agrees with the calculations of the CLSR.

*$\mathcal{PT}$-symmetric chaotic phase*: Figure 3 (right) shows the OTOC for the $\mathcal{PT}$-symmetric chaotic phase for $\lambda = (10^{-5}, 10^{-4})$, and fixed $K = 30$. As for the $\mathcal{PT}$-symmetric integrable phase, we find that the OTOC in the symmetric chaotic phase exhibits behavior similar to that of the Hermitian system in agreement with CLSR calculations.

*$\mathcal{PT}$-symmetry broken chaotic phase*: This is the only $\mathcal{PT}$-symmetry broken phase present in the phase diagram. In Fig. 3 (left), we show the OTOC for $\lambda = (10^{-3}, 0.005, 0.001)$, and fixed kicking strength $K = 30$. In Fig. 3 (right), the OTOC for $(\lambda, K) \in \{(0.006, 5.734), (0.01, 15)\}$ is shown. It displays two separate regimes of exponential growth. In early times, the growth is dominated by the chaotic nature of the Hamiltonian. In contrast, at late times, the growth is essentially because of the complexity of eigenvalues brought about by $\mathcal{PT}$-symmetry breaking, as we explain below.

In the $\mathcal{PT}$-symmetry broken phase, the eigenspectrum consists of complex eigenvalues, and thus under time evolution, the state $|\Psi\rangle$ no longer remains normalized (See Fig. 3). The late time growth of the OTOC (and state norm) depends on the eigenspectrum, in particular, on the maximum of the imaginary part of the eigenvalues of the Hamiltonian. Let us denote this maximum value by $\alpha$. Consider the late-time behavior of the state $|\Psi\rangle$

$$U(t \to \infty)|\Psi\rangle \propto e^{\alpha t}|\Psi\rangle, \tag{10}$$

Since, at late times, the evolution is dominated by the phase $e^{-iEt}$ from $E = E_r + i\alpha$, it follows that the late time norm $\mathcal{N}(t)$ obeys

$$\mathcal{N}(t \to \infty) \propto e^{2\alpha t}. \tag{11}$$

To consider the behavior of the OTOC, we explicitly write the commutation relation $-[\hat{p}(t), \hat{p}(0)]^2$ as

$$\hat{p}(0)\hat{p}(t)\hat{p}(t)\hat{p}(0) + \hat{p}(t)\hat{p}(0)\hat{p}(0)\hat{p}(t) - \hat{p}(t)\hat{p}(0)\hat{p}(t)\hat{p}(0) - \hat{p}(0)\hat{p}(t)\hat{p}(0)\hat{p}(t) \tag{12}$$

In the large $t$ limit, we can write the contribution from a single term as

$$\langle \Psi | \hat{p}(t) \hat{p}(0) \hat{p}(t) \hat{p}(0) | \Psi \rangle \propto e^{4\alpha t} \tag{13}$$

Therefore, the late time OTOC

$$C(t \to \infty) \propto e^{4\alpha t} \tag{14}$$

This behavior motivates a definition of a normalized OTOC

$$\tilde{C}(t) = e^{-4\alpha t} C(t). \tag{15}$$

The normalized OTOC, $\tilde{C}(t)$, is expected to exhibit exponential growth only due to the chaotic nature of the Hamiltonian. Since we are only performing calculations up to finite times, there is a contribution from sub-leading eigenvalues as well. Therefore, the exponent is obtained from an exponential fitting only for late times (see supplementary material). The normalized OTOC, then, only contains a growth due to the chaotic nature of the system (See Fig. 4) and can be used to extract a Lyapunov exponent. Numerically, in order to extract the exponent from $\tilde{C}(t)$, we determine the times, after which the exponential growth starts slowing down, and fit $\tilde{C}(t)$ from $t = 1$ up to these times to the function $ae^{2\gamma(t-1)}$ to find the parameter $\gamma$. Numerical overflows prevent a calculation of the Lyapunov exponent if it has a large value. We, thus, have to confine ourselves to a calculation of the exponent only in a region of the phase where $\mathcal{PT}$-symmetry is broken weakly.

In Figure 6, we show the Lyapunov exponent as well as the corresponding CLSR across the transition $T_2$ for varying $\hbar_{\text{eff}}$. We find that for all values of $\hbar_{\text{eff}}$, the Lyapunov exponent displays a peak. It is important to note that the peak does not appear at the transition $T_2$ but instead exists inside the broken phase of the $\mathcal{PT}$ symmetry. A further investigation of the nature of the peak in the Lyapunov exponent profile (See Sec. 5) is required. We also note that in a calculation performed on the same system as ours [55], a different definition of the OTOC has been used, one involving both the operators $\theta$ and $p$. The OTOC defined this way displays polynomial growth in time in the $\mathcal{PT}$ symmetry broken regime and not an exponential one, as we observe.

## 4.3 Parameter dependence

We summarize here the behavior of the CLSR and OTOC as we take three particular limits of interest.

- $N \to \infty$: This is the limit we take to cover the ring of allowed eigenvalues of $\hat{p}$ densely and approach the continuum limit. The CLSR and LE plots show little to no noticeable shift when increasing the value of $N$, as can be seen from Fig. 7. The phase transitions seen on the CLSR become sharper but still occur roughly at the same value of $\lambda$ with a slight drift.

- $\lambda \to 0$: This limit signifies the Hermitian limit of the problem where only two regimes have been identified with the CLSR values 0.5 and 0.57 corresponding to the integrable and chaotic regimes, respectively. These correspond to the PT-symmetric integrable regime and PT-symmetric chaotic regime for the general ($\lambda \neq 0$) case. It can be seen from Fig. 6 that the value of $K$ for the transition between the two phases in the Hermitian limit persists even for small values of $\lambda$ for finite values of $\hbar$.

- $\hbar \to 0$: This limit maps the quantum problem to its classical counterpart where again, there exists a phase transition from the integrable phase to a chaotic phase. As can be seen in Fig. 6, as $\hbar \to 0$, the transition from integrable-PT symmetric to chaotic-PT

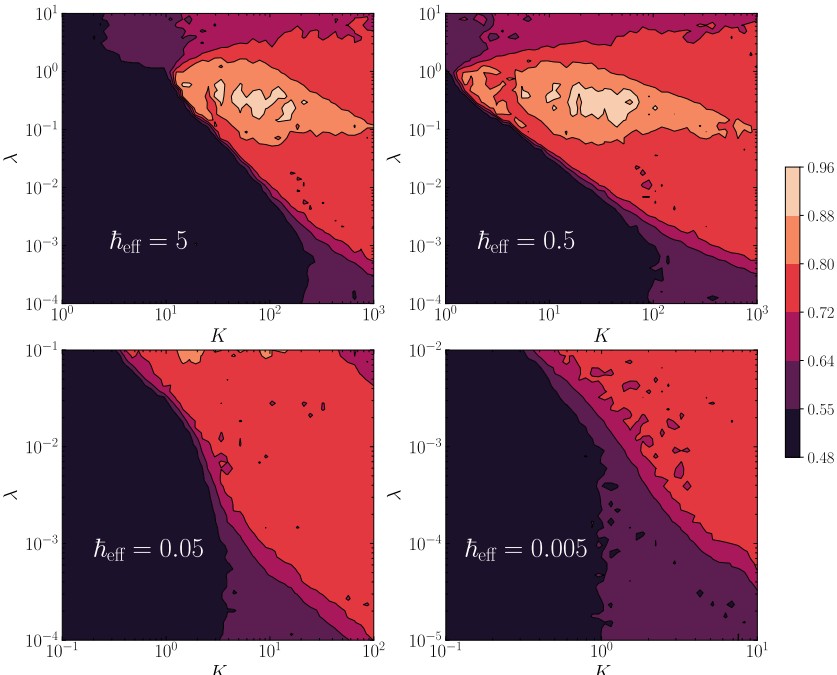

Figure 5: The above figure shows a section of the CLSR contour map for different values of $\hbar_{\text{eff}}$. It can be observed that the transition T* moves closer to the integrable → chaotic transition seen in the classical model for which $K \sim 1$ as $\hbar_{\text{eff}} \to 0$. Furthermore, the PT-symmetric chaotic regime is observed to shrink in size with a decrease in $\hbar_{\text{eff}}$

symmetric is happening at $K \approx 1$, which is where the classical transition occurs [56]. This observation is also in agreement with the transition observed for the Hermitian quantum kicked rotor [57] as $\hbar \to 0$. Further, we observe the shrinking of the chaotic-PT symmetric regime as $\hbar \to 0$. From the Hermitian case, we know that the chaotic-PT symmetric regime will always exist on the $\lambda = 0$ line. However, we postulate that in the limit as $\hbar \to 0$, the only phase present in the $\lambda \neq 0$ space will be the chaotic-PT unbroken phase, i.e, an infinitesimal value of $\lambda$ will be sufficient to break PT symmetry, leading to chaotic nature in this limit.

# 5 Conclusions

We have performed a comprehensive numerical study of the non-Hermitian kicked rotor demonstrating, for the first time (to our knowledge), the presence of integrability and $\mathcal{PT}$ symmetry breaking transitions in a driven quantum model. The model we have studied exhibits three phases, i) A $\mathcal{PT}$ symmetric integrable phase, ii) A $\mathcal{PT}$ symmetric chaotic phase and iii) A $\mathcal{PT}$ symmetry broken chaotic phase, thus demonstrating that $\mathcal{PT}$ symmetry breaking is a sufficient condition for the onset of chaos. We have characterized these phases and the transitions between them by calculating the complex level spacing ratio (CLSR) and the out of time ordered correlator (OTOC).

The CLSR phase diagram (see Fig. 2) elucidates several properties of the PTKR model and the intricate relation of $\mathcal{PT}$-symmetry breaking and chaos. Foremost, it shows that the CLSR is a viable diagnostic to differentiate the several phases in our system for both the non-Hermitian and Hermitian cases. One of the main observations is the absence of an integrable,

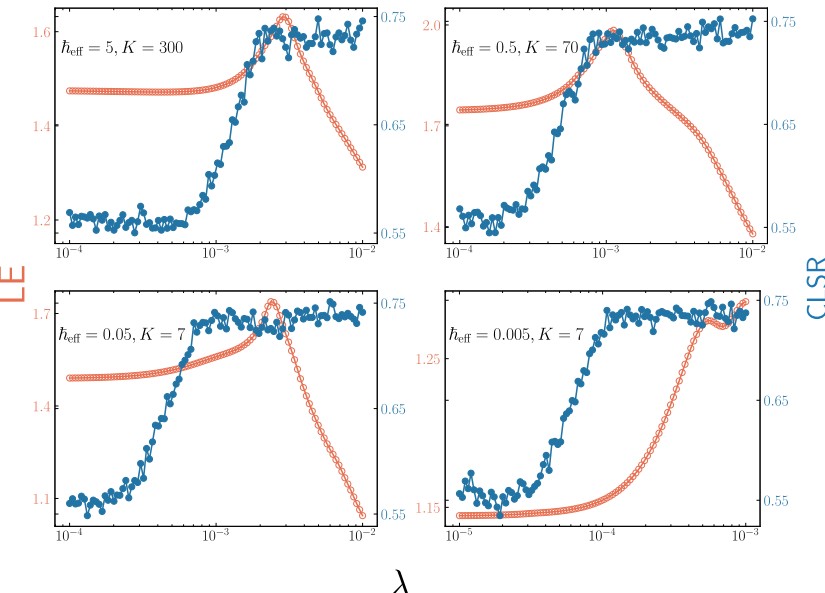

Figure 6: The above figure provides a comparison between the observed PT-symmetric chaotic → PT-symmetry broken chaotic regime transition probed by the CLSR and the accompanying LE values for the same set of parameters for $K$ and $\lambda$. These have been plotted for various values of $\hbar_{\mathrm{eff}}$ that span three orders of magnitude. We observe that this transition observed by the CLSR is accompanied by a peak in the LE. Note that different values of $K$ have only been used to ensure the transition is properly captured, since, as previously stated, the $\mathcal{PT}$-symmetric chaotic regime shrinks as $\hbar_{\mathrm{eff}} \to 0$.

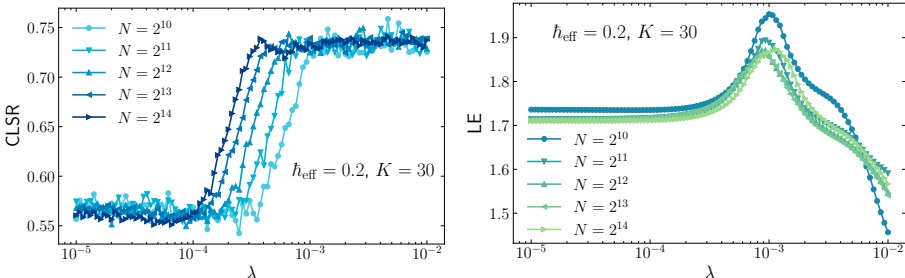

Figure 7: The above figure plots the transition from the PT-symmetric integrable regime to the PT-symmetric chaotic regime for different system sizes that increase by a factor of 2. $\hbar = 0.2$ and $K = 30$ for all the plots. It can be seen that the transition, as obtained from the CLSR gets sharper as the system size ($N$) increases. There is a slight shift in the value of $\lambda$ at which the transition occurs and this shift appears to decrease with successively increasing values of $N$. The peak in the LE also appears to remain at a nearly fixed value of $\lambda$ as the system size is varied.

$\mathcal{PT}$ symmetric phase in the phase diagram as determined from the complex level spacing ratio $\langle r \rangle$, which implies the sufficiency of $\mathcal{PT}$ symmetry breaking for the setting in of chaos. We also made a couple of other interesting observations, which, while not very clearly understood at this stage, motivated further work. The first is that a transition from between the $\mathcal{PT}$ integrable and chaotic phase can be effected by varying only the non-Hermiticicy parameter $\lambda$. This is quite intriguing since, naively, one might expect that such a transition would necessarily require varying the kicking strength, such as in the purely Hermitian system. The second observation is the appearance of a peak in the Lyapunov exponent as a function of the non-Hermiticity parameter $\lambda$ inside the $\mathcal{PT}$ symmetry broken phase close to the transition. A broad peak in the Lyapunov exponent has been observed in a Hermitian Bose-Hubbard in the vicinity of a quantum phase transition [32] but further investigation is required to determine whether or not there is any connection between the aforementioned observation and ours.

We obtain the Lyapunov exponent from a calculation of the OTOC. We show that the early time growth of the OTOC can be used to distinguish between the three phases we observe. In particular, once we define a normalized version of the OTOC to eliminate the effects of the growth in time due to the complex nature of eigenvalues, it can be employed to detect the transition from the $\mathcal{PT}$-symmetric chaotic phase to the $\mathcal{PT}$-symmetry broken chaotic phase. We also observe that the reality of the eigenvalues in the above two regimes does not display a sharp disappearance at the transition, so the origin of the sharp jump in the CLSR, which indicates the transition between the two phases, needs further investigation.

## Acknowledgements

H.S. would like to thank Aranya Bhattacharya for useful discussions. SM thanks the DST, Govt. of India for support.

## A  Numerics for the level spacing calculations

This section describes the computational schemes employed along with the necessary theoretical motivation.

### A.1  Modifying the Hamiltonian

The Hamiltonian of a $\mathcal{PT}$ symmetric kicked rotor is conventionally chosen to have the following form for the potential $V(\theta)$.

$$V(\theta) = K(\cos\theta + i\lambda\sin\theta) \tag{16}$$

However, in order to be able to simultaneously study the effect of varying the kicking strength $K$ and the non-Hermiticity parameter $\lambda$, the above form is not a good choice. For $\lambda \gg 1$, we see that $\lambda$ not only tunes the non-Hermiticicy parameter but also controls the overall 'magnitude' of the $V(\theta)$ term and so controls the kicking strength as well. To avoid this, we 'normalize' the term inside and use the following definition for $V(\theta)$.

$$V(\theta) = \frac{K}{\sqrt{1+\lambda^2}}(\cos\theta + i\lambda\sin\theta) \tag{17}$$

### A.2  Accounting for degeneracies due to symmetries

While calculating the energy level spacing distribution from the eigenvalue spectrum of a Hamiltonian, it is very important that one accounts for degeneracies arising from symmetries

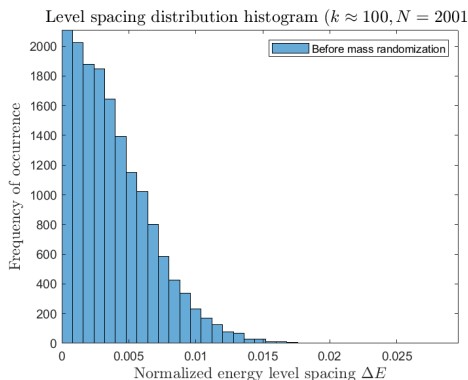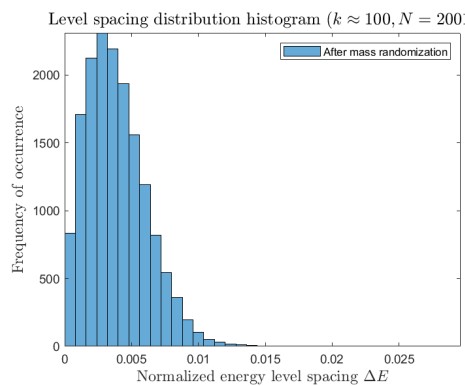

Figure 8: The level spacing distribution for $k \simeq 100$. It can be seen that due to degeneracies arising from symmetries or near-symmetries, the distribution does not match the GOE form (left panel). The distribution matches the GOE form after introducing a random small perturbation of $10^{-3}m$ (right panel).

of the Hamiltonian. Hence, it is a common practice to carry out the level spacing calculations on a subset of the set of eigenvalues without repeating degenerate eigenvalues. The Hermitian kicked rotor Hamiltonian has parity $\mathcal{P}$ as one of its symmetries. To achieve such a subset, we block diagonalize the Hamiltonian using a basis that diagonalizes the group of symmetries of the Hamiltonian. Then, we perform the level spacing calculations over one of the boxes of the Hamiltonian.

In our case, defining the Floquet operator and retrieving the eigenvalues results in a large number of unidentified symmetries and near symmetries, causing a peak to exist near $\Delta E$ (level spacing) $= 0$ in the distribution, as can be seen in figure 8. To improve the statistics and get rid of such near symmetries, we modify the kinetic energy term in the Hamiltonian. When written in the angular momentum basis, the kinetic energy is diagonal with its diagonal elements being $-\dfrac{\hbar^2 l^2}{2m}$ where $l$ is the angular momentum eigenvalue and $m$ the moment of inertia. To break such symmetries, we add randomness to the masses of each diagonal term individually by redefining the diagonal elements as $-\dfrac{\hbar^2 l^2}{2(m + \delta m)}$. For our calculations, $m = 1$ and the $\delta m$'s are chosen randomly from a uniform distribution over the interval $[0, 10^{-3}]$ for each diagonal element. This has the desired effect while still preserving the transition from the integrable to the chaotic phase. Figure 8 provides an example of this in the Hermitian case. The Hermitian version of the PTKR model has $\mathcal{P}$ and $\mathcal{PT}$ as symmetries of the system. The plot is of the Wigner-Dyson level spacing distribution.

## A.3 Unfolding procedure

Whether using real or complex eigenvalues, the unfolding process is crucial to getting a faithful representation of the energy level spacing distribution. The energy level spacing has the dimensions of energy. Wigner's analysis of the level spacing distributions involved calculations over a large ensemble of $2 \times 2$ matrices with particular symmetries. Suppose that this is also true for the distribution obtained from an $N \times N$ random matrix with the same symmetries over its spectrum. In this case, it must be that the level spacing in different parts of the spectrum is not related or weakly related. Consequently, in order to compare level spacing across distinct sections of the spectrum, it becomes necessary to work with dimensionless quantities. These quantities can be obtained by dividing the level spacing by the mean level spacing in a small vicinity surrounding them. Such a procedure is known as unfolding.

For a real spectrum, the mean is simply obtained by averaging the level spacing about that eigenvalue for a fixed radius of points. For a complex spectrum, we define $\frac{1}{\sqrt{\rho_k}}$ as the mean level spacing around an eigenvalues $\xi$. Here $\rho_k$ is defined as follows.

$$\rho_k = \frac{3n}{\pi\left(r_{k,n-1}^2 + r_{k,n}^2 + r_{k,n+1}^2\right)} \tag{18}$$

Above, $n$ is predefined depending on our Hilbert space dimensions $N$. In our calculations, $n = 10$ appears to work well for $N = 2001$. Note that real or complex level spacing ratios do not need to undergo unfolding procedures since they already involve averaging over dimensionless quantities.

### A.4  Averaging over intervals of $K$

In order to make the level spacing ratios and distributions robust to any kind of special high symmetry $K$ values and to obtain better statistics at a much lower computational cost, we average over intervals of $K$. For kicking strengths far away from the transition points, the behavior of the energy level spacing ratio and distribution does not change much at all. Further, since the folding of the eigenvalue spectrum depends greatly on the kicking strength, there can be resonances in the quantum mechanical model reminiscent of those in the classical model. This can cause the suppression of the chaotic nature of the system. Also, rather than working with a higher dimension Hilbert space, which requires $\mathcal{O}(N^2)$ time, we can simply combine the eigenvalues of nearby kicking strengths, which only takes $\mathcal{O}(N)$ time. Note that because the magnitude of eigenvalues can differ for different kicking strengths, we normalize the spectra for separate kicking strengths so that it lies in the interval $[0, 1]$ and then concatenate them. This technique has not been employed for the calculations that yield color plots. However, this is very effective if one wishes to calculate the CLSR or RLSR in a particular neighborhood away from the transitions.

## B  Calculation on random matrix ensembles and new CLSR

In our work, we show the agreement of the level spacing ratio obtained from our model and that from calculations over random matrices with the same symmetries as those of the model. Some of these values have been calculated previously as well [**?**]. We perform random matrix calculations for complex universality classes GinUE, GinOE, and AI$^\dagger$ [49]. Here, we highlight the method used to perform such calculations.

First, a complex matrix with $2N^2$ real entries is defined, giving complex random matrices of size $N \times N$. Every real entry is independently and identically chosen from a normal distribution of mean 0 and standard deviation 1. Any further symmetry constraints are added as follows: Let the symmetry group of these constraints be $\mathcal{A}$, then all the elements of $\mathcal{A}$ are made to act on copies of the same random matrix, and all of the resultant matrices are added. Next, the eigenvalues of the matrix thus obtained are numerically calculated, and the CLSR is found. This process is repeated enough times so as to ensure that the standard deviation of all the measures is sufficiently small compared to the mean.

The complex level spacing ratio can also be defined using angles, which is the other definition provided in [50]. This definition has not been adopted by us since it fails for a real eigen-spectrum where $\theta$ becomes ill-defined. However, we perform random matrix calculations for $-\langle\cos\theta\rangle$ as well as described below.

Table 2: The above table consists of data gathered on the complex level spacing ratio for some complex universality classes (GinUE, GinOE, and AI$^\dagger$); bracket contains must be read as (mean value, standard deviation).

| Universality class | $\langle r \rangle$ | $-\langle \cos\theta \rangle$ |
|---|---|---|
| A/GinUE | (0.73809, 0.00207) | (0.23297, 0.0093) |
| AI$^\dagger$ | (0.7231, 0.0027) | (0.18777, 0.0077) |

Table 3: The above table consists of data gathered on the CLSR for the $\mathcal{PT}$ symmetry broken chaotic regime and its comparison with CLSR in the complex universality classes GinOE and its subset of matrices that commute with $\mathcal{PT}$, bracket contents must be read as (mean value, standard deviation).

| Universality class | CLSR |
|---|---|
| Phase diagram | (0.735) |
| GinOE | (0.73809, 0.0048) |
| $\mathcal{PT}$-symmetric matrix | (0.73942, 0.0054) |

## C  $\mathcal{PT}$-symmetry broken phases

The quasienergy of the PTKR model is complex i.e. $\epsilon = \epsilon_r + i\epsilon_i$. When the maximum value of the imaginary part (denoted by $\alpha$) exceeds a certain threshold value $\alpha_c$, we assume the $\mathcal{PT}$-symmetry to be broken. The yellow shaded region is the $\mathcal{PT}$ symmetry broken region. In parts of the region near the transitions, we obtain a CLSR close to 0.739. This is the value of the CLSR one obtains for the GinOE universality class, to which the Hamiltonian belongs. This is due to the fact that the Hamiltonian is not bound by Hermiticity, and it commutes with an anti-unitary operator (namely $\mathcal{PT}$). In the table below, we show calculations of the CLSR on random matrices in the GinOE class. Additionally, we also perform calculations on matrices within GinOE that commute with $\mathcal{PT}$. These are in agreement with the value of the CLSR we get in our phase diagram.

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
