# Peer review of "Quantum chaos in PT symmetric quantum systems"

_SciPost Physics_

## Round 2 · Referee Report · Anonymous (Referee 1) · 2025-1-8

Strengths

The authors study the interplay of nonhermiticity and chaos in the nonhermitian kicked rotor.
Using numerical methods they show that the breaking of PT symmetry engenders chaos.
They use the concept of complex level spacing ratios to address the transition to chaos.
They find that PT breaking is always accompanied by chaos.
This is an interesting result.

Weaknesses

However, 1. I found the paper hard to read and not well written. 2.There is not enough discussion of the state of the art of the field and physics of the model, which would have been very helpful to the reader to place the results in context. The paper fails to give a sense of what results are already known in the field as such, making it is hard to see which results are new.
3. There are lots of typos, missing references and important quantities are not described in the manuscript.

Report

In my view, despite the interesting result, the paper needs a bit of a makeover and the following questions/points need to be addressed before any recommendation for publication can be made.

Requested changes

Here are a few points:

Please briefly elaborate on Wigner’s surmise that is mentioned in the first paragraph.

Page 2: the authors motivate the paper as an extension of previous works. It would be useful to describe what the previous results are and highlight the contribution of the current work.

Page3: Define z_K, is K the strength ? Important as this forms the backbone of the work. How does one interpret this definition in a manner akin to normal level statistics for Hermitian systems?

More info on the values for CLSR for Poisson and GOE…one does not get any sense of why these values are higher for PT symmetric matrices compared to the case of real symmetrical matrices.

In Fig1, left caption: mention lambda=0, in the right figure, though RLSR is mentioned in the caption…there is no RLSR plotted

Pg.3 There are two computational techniques to improve the statistics in our calculation. The first is to replace m in Eq. (1) by m + ∆mp, where ∆mp is a small random number selected independently for each p.

Please indicate what this p corresponds to. It is discussed in the appendix and it would make sense to have a part of those statements repeated in the main body of the paper. I do not see the impact of App A.1 where the coupling strength is renormalized …as the authors really do not study the regime \lambda/K >>1 in this work. So is this necessary ?

Page 5. However, in this limit, Eq. 7 does not yield the standard real level spacing ratio (RLSR) for a real spectrum.
Given the plots in Fig.1, where the CLSR approaches the RSLR values for GOE etc for the Hermitian case, could the authors try and explain why the CLSR as given by Eq. 7 give different values when there is nonhermiticity ? If z_K only involves the Floquet spectrum which is real in the PT symmetric case, I do not understand what leads to these differences. In absence of any discussion on how z_K is defined, it is very opaque. A brief explanation of why one averages over the kicking strength will be useful. I think the authors should elaborate on these points.
A few plots of the quasienergy spectrum in the appendices would be illuminating.

Page 6. Regarding the different lines in Fig2, T1, T2 etc… is the boundary where PT symmetry breaking is observed a numerically obtained one by considering the Floquet eigenvalues or is there some analytical reasoning which can explain this ? Have the authors looked at possible similarity transformations of the Hamiltonian which gives rise to a pseudo-hermitian Hamiltonian, as this would help indicate the boundary between PT symmetric and PT broken regimes. This is something that one often does in non-hermitian problems (cf. Bender books and papers that the authors have referenced) Such a calculation will eminently help improve the clarity of the paper. Or is there some simplifcation that can be done in the limit \hbar-> 0 ? I would like the authors to address this.

Page 6: Caption: Right: The absolute value of the maximum imagi- nary part of an energy eigenvalue, α, across the transition Main panel: T1 Inset T2, calculated for N = 4095. It can be seen that while α shows an abrupt change along T2, it seems to increase smoothly along T1. The CLSR on the other hand, shows an abrupt transition along both T1 and T2.

This is very confusing as, there is a rather sharp change (within numerical finite size effects) in alpha along the T1 line as PT symmetry is broken. One expects alpha to show a non-analytic behaviour as the system crosses an exceptional point. In the inset, along T2, there is only one point which is off the smooth straight line behaviour of alpha. The authors seem to be drawing the opposite conclusion. I would appreciate a clarification of this.

Furthermore, in the inset for alpha what is varied across the x-axis ? The caption mentions that it is the variation along the T2 line, but there both lambda and K vary…so this is confusing. Additionally, the scales in the inset do not conform to the color map axes.

Table1 caption …please mention that this is valid for the PT symmetric case only. In fact, I would recommend that the authors discuss the complex random matrix ensembles discussed in the appendix here and link it to the numerical results that obtain for the CLSR. Such a discussion would be very useful to contextualize the numerical results.

Sec 4.1

In Fig. 1, we show the mean RLSR as well as the CLSR with varying values of the kicking strength K, for ħh = 0.2 and system-size N = 8005. We notice that the transition points are reasonably independent of the value of N. Thus, in what follows, the system size is chosen to be large enough so that all quantities are well converged. We find that both the CLSR and RLSR display a transition at the same value of K.

In Fig.1, why does the deviation from Poisson statistics happens at quite different K values in the
Hermitian case? Is there any understanding of this that the authors can provide ?

In Figs.3 and 4. I think the caption misstates which figures correspond to T1 and T2. This again confuses the reader.

In all the parameter regimes shown in the normalized OTOC, the long time behaviour approaches a constant…contrary to the authors; statement, this seems to be true irrespective of whether it is chaotic/PT broken or not. Could this authors discuss this more ?

In Fig.5, it would make for easier reading if all the axes had the same ranges to see the points made by the authors.

The main result of the paper is that PT breaking is always accompanied by chaos. Can the authors address why there is no PT broken phase without chaos ? Could this plausibly be a feature of the particular model studied ? In the random matrix ensembles that the authors allude to for complex matrices and eigenvalue structures, is there a possibility to have integrability to chaos transitions or does RMT prohibit these ? As this is the principal result of the work, the authors should present some arguments, atleast heuristic to bolster their results.

Page 9: Given that non-unitary evolutions are known to not preserve state norms, I recommend only plotting the normalized OTOC in the main body of the paper and move Fig 3 with the Norm and OTOC to the supplemental material.

Page 13.
…PT symmetric phase in the phase diagram as determined from the complex level spacing ratio 〈r〉, which implies the sufficiency of PT symmetry breaking for the setting in of chaos. We also made a couple of other interesting observations, which, while not very clearly understood at this stage, motivated further work.

I think the authors mean that an integrable PT broken phase is not present in this model.

Page 14:
typo
formafter

Page 15…missing reference
Some of these values have been calculated previously as well [?].

In Table 2, GinOE is missing …

Table 3…Is this table indicating that PT broken and PT symmetric chaotic regimes are described by the same RMT ? If yes, this is very curious as in quantum problems with no chaos, PT broken regimes have very different behaviours of observables as compared to
PT symmetric regimes. So if the onset of chaos blurs any difference, this would be a very important point to make. Are there other observables/quantities that one can numerical obtain for the PTKR that explore this further ? The authors should comment on this.

Appendix C PGE 16
The yellow shaded region is the PT symmetry broken region.
…which figure are the authors referring to ? Cannot find yellow regions in the figures presented.

Recommendation

Ask for major revision

  • validity: high
  • significance: high
  • originality: good
  • clarity: low
  • formatting: good
  • grammar: good

Author:  Subroto Mukerjee  on 2025-07-25  [id 5680]

(in reply to Report 1 on 2025-01-08)

Please see the attached file.

Attachment:

Response_to_the_referees_KcDiG4C.pdf

---

## Round 2 · Referee Report · Anonymous (Referee 2) · 2025-4-7

Strengths

  1. Well-motivated
  2. Clearly-written
  3. Topic is fundamentally important
  4. Detailed study

Weaknesses

None

Report

In the work, the authors study a problem of fundamental importance, namely quantum chaos in PT-symmetric quantum systems. PT-symmetric quantum systems are of great fundamental importance as they can feature real eigenvalue spectrum, despite the Hamiltonian being non-Hermitian. In the Introduction, authors present a comprehensive discussion on the current status and understanding of quantum chaos in Hermitian systems and motivate their study in PT-symmetric non-Hermitian system. Through their detailed study, the authors report three distinct phases; (1) a PT-symmetric integrable phase, (2) a PT-symmetric chaotic phase, and (3) a PT-symmetry broken chaotic phase. Their numerical results are sufficiently clear and convincing to support the claims. In the the Conclusion section, the authors also point out the open questions based on their findings. Although the topic is technical, the authors presented their results in a coherent fashion that can be followed (at least qualitatively) by non-experts and non-practitioners. This paper should be published in SciPost. I have only a minor suggestion for the authors to consider before the publication of this article.

Requested changes

In the sixth paragraph of the Introduction, the authors discuss applicability of non-Hermitian Hamiltonians in various other context, such as topological phases of matter. In this context, I request them to include effects of e-e interactions, quantum criticality, and Lorentz symmetry in NH systems, which have been studied in the recent past. Some of the early works in this field include

Physical Review Letters 132, 116503 (2024), Communications Physics 7, 169 (2024), Journal of High Energy Phys. 01, 143 (2024), SciPost Phys. 18, 073 (2025)

This works are particularly relevant in the context of the present study as they also consider effects of interactions in NH systems with real eigenvalue spectrum (admittedly in different setups).

Recommendation

Ask for minor revision

  • validity: high
  • significance: top
  • originality: high
  • clarity: good
  • formatting: excellent
  • grammar: perfect

Author:  Subroto Mukerjee  on 2025-07-25  [id 5679]

(in reply to Report 2 on 2025-04-07)

Please look at the attached file

Attachment:

Response_to_the_referees.pdf

---

## Editorial Decision

resubmitted